# Classification Importance of Seed Morphology and Insights on Large-Scale Climate-Driven Strophiole Size Changes in the Iberian Endemic Chasmophytic Genus *Petrocoptis* (Caryophyllaceae)

**DOI:** 10.3390/plants13223208

**Published:** 2024-11-15

**Authors:** Jorge Calvo-Yuste, Ángela Lis Ruiz-Rodríguez, Brais Hermosilla, Agustí Agut, María Montserrat Martínez-Ortega, Pablo Tejero

**Affiliations:** 1Área de Botánica, Universidad de Salamanca, 37007 Salamanca, Spain; angelalis00@usal.es (Á.L.R.-R.); mmo@usal.es (M.M.M.-O.); 2Herbario y Biobanco de ADN Vegetal, Universidad de Salamanca, 37007 Salamanca, Spain; 3Banco de Germoplasma Vegetal del Jardín Botánico de Olarizu, 01006 Vitoria-Gasteiz, Spain; brais.hermosilla@gmail.com (B.H.); aagut@vitoria-gasteiz.org (A.A.); 4Herbario JACA, Instituto Pirenaico de Ecología (IPE-CSIC), 22700 Jaca, Spain; 5Aranzadi Sociedad de Ciencias, 20014 Donostia-San Sebastián, Spain

**Keywords:** chasmophyte, cliff environment, Iberian Peninsula, LMEs, local adaptation, machine learning, morphometrics, random forest, water availability

## Abstract

Recruitment poses significant challenges for narrow endemic plant species inhabiting extreme environments like vertical cliffs. Investigating seed traits in these plants is crucial for understanding the adaptive properties of chasmophytes. Focusing on the Iberian endemic genus *Petrocoptis* A. Braun ex Endl., a strophiole-bearing Caryophyllaceae, this study explored the relationships between seed traits and climatic variables, aiming to shed light on the strophiole’s biological role and assess its classificatory power. We analysed 2773 seeds (557 individuals) from 84 populations spanning the genus’ entire distribution range. Employing cluster and machine learning algorithms, we delineated well-defined morphogroups based on seed traits and evaluated their recognizability. Linear mixed-effects models were utilized to investigate the relationship between climate predictors and strophiole area, seed area and the ratio between both. The combination of seed morphometric traits allows the division of the genus into three well-defined morphogroups. The subsequent validation of the algorithm allowed 87% of the seeds to be correctly classified. Part of the intra- and interpopulation variability found in strophiole raw and relative size could be explained by average annual rainfall and average annual maximum temperature. Strophiole size in *Petrocoptis* could have been potentially driven by adaptation to local climates through the investment of more resources in the production of bigger strophioles to increase the hydration ability of the seed in dry and warm climates. This reinforces the idea of the strophiole being involved in seed water uptake and germination regulation in *Petrocoptis*. Similar relationships have not been previously reported for strophioles or other analogous structures in Angiosperms.

## 1. Introduction

The singularity, remarkable diversity and ecology of rock-dwelling plants have piqued the interest of the scientific community at least since the middle of the last century [1,2]. Given the abundance of endemic and/or threatened rupicolous taxa and the particularities of rocky habitats [3,4,5], an increasing number of authors have contributed to enlarging our knowledge about plant life in rocky habitats worldwide (e.g., [6,7,8]) and in Mediterranean area (e.g., [9,10,11]). Rocky outcrops are fragmented environments where a great diversity of specialist plant taxa—often with narrow distribution areas—occur. These outcrops are proposed to be a result of the climatic and topographic changes throughout the Cenozoic [2,12,13], when the Alpine orogeny led to the formation of most Indo-European mountain chains which, together with the existing ones, have been shaped by erosive forces since then. These processes were intensified during the Pleistocene climatic oscillations and resulted in a diversity of rocky microenvironments, which played a key role as refugia for the flora both in the glacial and interglacial periods [14,15]. Consequently, rocky habitats would have acted as reservoirs of relict biodiversity [16] and promoted diversification of rock specialist plant lineages [17,18]. In this context, some widely distributed plant lineages probably underwent transitions towards small ecologically and taxonomically isolated populations and/or taxa [19,20,21]. The inherent disjunction of rocky habitats leads to diversification processes, and therefore, a geographic pattern may be found across the range of taxonomically coherent accepted species, in which different groups—either genetic or morphological—might be identified in different regions [22,23]. Thus, those species may be composed of a mosaic of biological units with a variable degree of isolation that are potentially responding to different adaptive pressures. Given this uncertainty, the study of rupicolous genera is needed to shed light on their evolutionary history, understand their adaptation strategies and promote their conservation.

The Iberian Peninsula, as part of the Mediterranean biodiversity hotspot, hosts a high richness of rupicolous taxa and, together with its great geological and topographical complexity, provides an ideal study location for the abovementioned purpose. Due to its complex palaeoclimatic and palaeogeographic history and to the huge variety of rocky environments available (e.g., [24,25,26,27]), the Iberian Peninsula lodges a high diversity of rock-dwelling taxa, many of which are endemic [5]. Among the 30 largest genera of the Iberian endemic flora [28], the following ones stand out as eminently rupicolous: *Armeria* Willd. (Plumbaginaceae, ca. 55% of the species present in the Iberian Peninsula are rupicolous, based on estimations according to our own unpublished data), *Teucrium* L. (Lamiaceae, ca. 51%), *Saxifraga* L. (Saxifragaceae, ca. 90%), *Erodium* L’Hér. (Geraniaceae, ca. 61%), *Campanula* L. (Campanulaceae, ca. 52%), *Scrophularia* L. (Scrophulariaceae, ca. 52%), *Hieracium* L. (Asteraceae, ca. 62%) and *Chaenorhinum* (DC.) Rchb. (Plantaginaceae, ca. 53%). Accordingly, rocky outcrops of the Iberian Peninsula would have acted as important centres of diversity for many genera or infrageneric groups (e.g.,: [10,11,29,30,31,32,33,34]), as well as a centre of origin for others such as the endemic Iberian rock-dwelling genera *Rivasmartinezia* Fern; Prieto & Cires (Apiaceae), *Dethawia* Endl. (Apiaceae), *Phalacrocarpum* (DC.) Willk. (Asteraceae) and *Petrocoptis* A. Braun ex Endl. (Caryophyllaceae), our study model.

*Petrocoptis* is an Iberian-endemic genus that comprises between 4 and 12 chasmophytic species, depending on the authors (see Section 4). It is phylogenetically closely related to *Silene* L. and *Agrostemma* L. [35,36,37], from which it differs, among other traits, by the presence of a strophiole, a characteristic small tuft of hairs by the hilum of its seed [38]. Strophiole morphology has also been used as a taxonomic character to distinguish species within the genus [39]. Nearly all *Petrocoptis* species are included in conservation lists at European and Spanish levels—either national or regional—due to their restricted ecology and the small distribution area of the taxa [40,41]. However, the different taxonomic treatments available for the genus sometimes differ strongly among them, which interferes with the appropriate evaluation of the conservation status of the taxa concerned. Consequently, it is essential to resolve the taxonomic uncertainty that affects *Petrocoptis*. In this respect, a phylogenomic analysis of the whole genus is being performed at the moment by our team, and thus, a taxonomic objective is excluded from this work. Moreover, comprehensive knowledge of the genus evolution and its current fragmented distribution pattern linked to habitat specificity should be integrated into a meaningful taxonomic treatment.

Despite its restricted distribution range, *Petrocoptis* inhabits cliffs comprising a wide altitudinal range (from 20 to more than 2000 m a.s.l.) and climatic spectrum (temperate to Mediterranean macrobioclimates, following [42]). Under these broad conditions, individuals may be phenotypically plastic, expressing the optimal phenotype in different environments without genetic differentiation [43,44,45], or populations may differentiate genetically so as to become locally adapted [46,47,48]. Although plants are thought to be generally plastic [49], populations can differ greatly in their plasticity levels [50,51,52,53], especially when habitat heterogeneity and/or isolation limit gene flow among them [54]. In addition, it has been observed that plasticity can be facilitated or limited by climatic variables [55,56]. Additionally, it is probable that phenotypic plasticity and precipitation or temperature are correlated, with benign conditions favouring greater phenotypic plasticity [57,58] due to a wider range of morphological and physiological variations [59]. On the contrary, stressful conditions may place restrictions on phenotypic plasticity [60]. In the case of *Petrocoptis*, prior research showed high plasticity in germination throughout most of the species of the genus [61], but further research is needed to evaluate plasticity in seed morphology traits. All mentioned features make the genus *Petrocoptis* an ideal study case to understand plant specialization in rocky habitats.

The ecology of chasmophytic plants is relevant to understanding their past, present and future distribution. Although spatially open, a vertical cliff is a biologically closed community [2] because the space that separates one cliff from another is often inadequate for plant establishment. Therefore, range expansion is usually extremely difficult for plants living on vertical cliffs. In these habitats, the suitable holes on rock surfaces available for seedling establishment are very scarce and, when they originate, tend to be occupied by the seeds of species already present nearby. Consequently, medium- and long-distance dispersal is unfavoured, and population isolation is enhanced. Moreover, chasmophytes would disappear from a cliff without the presence of specialized traits that favour the dispersal and establishment of the progeny. Therefore, the seeds of chasmophyte plants are of primary research interest. Wind is believed to be the main dispersal agent for plants, but it is dependent on the lightness of the diaspore, either seeds or fruits [62,63]. However, there are cliff specialist plants with no wind-compatible syndrome. Some of them have a diaspore with a fatty-rich appendage, the elaiosome, which is widely thought to function as a nutritional reward for mutualistic ants that indirectly disperse the diaspores (i.e., myrmecochory [64]). This structure has been described in certain Iberian rupicolous plants [65], which has led to the associated presence of an elaiosome or other structures resembling it with the myrmecochory syndrome [66,67]. However, each case has to be investigated independently in search of empirical evidence.

An analogous structure, which has been traditionally used to distinguish *Petrocoptis* from other Caryophyllaceae genera, is the strophiole [68]. It is an outgrowth of the hilum region that restricts water movement into and out of the seed [69,70]. As stated, *Petrocoptis* strophiole is composed of a very conspicuous tuft of either cylindrical or claviform hairs of variable size [39,71], which have hygroscopic properties and become mucilaginous on wetting [72]. In addition to its function as a regulatory structure of the seed water balance, it has also been considered an adaptive trait related to seed dispersal. For example, refs. [67,73] have considered the strophiole as a structure related to myrmecochory. Alternatively, refs. [72,74] discussed that the hygroscopy of the strophiole facilitates adherence within the damp crevices where *Petrocoptis* lives, promoting a specialization to these habitats. An exhaustive analysis of seed traits, including the strophiole, is needed to fully understand the role that its particularly heavy seed plays in the dispersal and rock specialization of the genus *Petrocoptis*.

The aim of this study was to investigate the morphology of the seeds of *Petrocoptis* in order to (1) assess whether the measured morphometric seed traits can contribute to the description of seed morphogroups that hold taxonomic significance); (2) explore whether a relationship can be established between these traits and climate; and (3) try to obtain further clues on the biological role that the strophiole may play in *Petrocoptis*. Taking into account intrapopulation variability associated with phenotypic plasticity and a potential high interpopulation variability due to the high degree of population isolation, seed morphology may have taxonomic value. Additionally, a correlation between strophiole-related traits and climate can be expected, considering that this structure is suspected to play a key role in water uptake and regulation.

## 2. Results

### 2.1. Clustering and Recognisable Morphological Groups

Based on all nine analysed morphological traits, the grouping of population means into three clusters received strong support. A total of 16 out of the 27 indices calculated by the NbClust algorithm converged k = 3 as the optimal number of clusters. The remaining 11 indices showed much weaker support, with less than 4 of them agreeing on a different number of clusters (Appendix A). Despite the high inter-population variability, the variation between all clusters (between cluster sum of squares, BCSS) was higher than the variation within them (within-cluster sum of squares, WCSS): BCSS/WCSS = 1.364.

The first two principal components (PC1 and PC2) were the ones that best illustrated the separation among clusters, explaining 56.27% and 27.59% (83.86% in total) of the population variance, respectively (Figure 1). PC1 showed a high positive correlation with the variables related to the raw size of both the seed and the strophiole, while PC2 was positively correlated with the strophiole proportional size and seed roundness (Appendix A). As expected, the width and length variables of both seed and strophiole showed a high correlation with their respective areas. Therefore, they were discarded. The correlation coefficients between the remaining five traits did not exceed the value of 0.74 for any pair. Thus, seed area, seed roundness, strophiole area, strophiole roundness and strophiole relative size were retained for clustering analyses.

Cluster “a” (circles in Figure 1), characterized by high seed area, low strophiole area and low strophiole relative size, included populations from *Petrocoptis crassifolia* Rouy, *Petrocoptis montserratii* Fern.Casas (except for Riglos, RIGL) and Beranuy (BERA), identified as *Petrocoptis montsicciana* O.Bolòs & Rivas Mart Cluster “b” (triangles), consisting of individuals showing low seed area, low strophiole area and medium strophiole relative size, included all populations of *Petrocoptis grandiflora* Rothm., *Petrocoptis pseudoviscosa* Fern.Casas, *Petrocoptis pyrenaica* subsp. *pyrenaica* (Bergeret) A.Braun ex Walp., *Petrocoptis pyrenaica* subsp. *viscosa* (Rothm.) P. Monts. & Fern. Casas and *Petrocoptis pyrenaica* subsp. *glaucifolia* (Lag.) P. Monts. & Fern. Casas, except for the populations from El Pindal (PIND) and Potes (POTE) of the latter. Likewise, this cluster included the population from Teller (TELL), identified as *P. montsicciana*. Cluster “c” (squares), characterized by a medium seed area, a high strophiole area and a high strophiole relative size, included all the populations from *Petrocoptis guarensis* Fern.Casas, *Petrocoptis hispanica* (Willk.) Pau, *Petrocoptis pardoi* Pau, *P. montsicciana* (except those mentioned above) and the populations from Riglos (RIGL, *P. montserratii*), El Pindal (PIND, *P. pyrenaica* subsp. *glaucifolia*) and Potes (POTE, *P. pyrenaica* subsp. *glaucifolia*).

The subsequent one-way Welch’s ANOVA showed significant evidence of morphological differences among the three groups when combining seed area, strophiole area and strophiole relative size (Figure 2). Both seed and strophiole roundness showed few or no differences across groups. 

Furthermore, the random forest model displayed an overall accuracy of 0.8723, which means that ca. 87% of the seeds were correctly classified (Table 1). Despite group size inequality, the significant difference between the no-information rate and the overall accuracy outlined a good performance across groups. The balanced accuracy of classification for each group was also high and consistent (ca. 91% for group a, 89% for group b and 85% for group c). The traditional taxonomic classification selected as reference for the genus in 9 species or 11 taxa (including subspecies, [39]) showed worse predictive results: ca. 63% and 48% of correct classifications, respectively (Appendix A). The more synthetic taxonomic proposals by Walters [38], which classified the genus in five species or seven taxa, and Mayol & Rosselló [71,75], with four species or seven taxa, obtained slightly better predictive results: ca. 68%, 54%, 77% and 69% correct classifications, respectively. Nonetheless, their classification showed worse predictive results than our cluster arrangement.

In line with previous ANOVA results, the variables with the highest classifying importance under the random forest model assumptions turned out to be the seed area, the strophiole area and the strophiole relative size. When combining these morphological traits, the support vector machine algorithm created a space of classification probabilities of a random seed in a given group (Figure 3). Thus, the highest probability of classifying a seed as part of cluster “a” (>0.6) was obtained when the seed area exceeded 2 mm^2^ and the strophiole relative size was below the threshold of 0.5. In the case of cluster “b”, any seed with a seed area of <1.75 mm^2^ and a strophiole area <1 mm^2^ was classified as such with a high probability (>0.8). Last, the highest values of classificatory probability within cluster “c” (>0.7) were obtained when the strophiole relative size exceeded the threshold of 1.

### 2.2. Climate—Seed Morphology Relationship

The best-fitted linear mixed effects model showed that the strophiole area of *Petrocoptis* seeds is sensitive to climate. Average annual rainfall was the climatic variable that explained most of the variability of the strophiole area (ΔAIC = 6.427; Table 2) and showed a negative correlation with it (Figure 4, Table 3). The average annual maximum temperature was the other most explanatory variable within the best model (ΔAIC = 4.517) and showed a positive correlation with the strophiole area. Furthermore, the relative size of the strophiole (i.e., strophiole area/seed area), a variable calculated to control the seed area effect on the strophiole, was consistent with the raw size model. Again, average annual rainfall explained most of the variability (ΔAIC = 7.334) and was negatively correlated. Temperature remained a driver of strophiole relative size (average annual maximum temperature ΔAIC = 7.019) and was positively correlated. On the other hand, the best-fitted and most parsimonious seed area model did not include any of the considered climatic variables and suggested that there is no evidence of a climate-driven seed size variation in our dataset. Exploring the variable importance through RMSE loss function, nested population and individual levels was the most important variable of both strophiole area and strophiole relative size models (ΔRMSE = 0.162 and 0.119, respectively; Figure 5), followed by average annual rainfall (ΔRMSE = 0.117 and 0.090) and average annual maximum temperature (ΔRMSE = 0.124 and 0.092).

When implementing the grouping factor within the best climate models, it turned out to be the most fixed effect that collected the greatest amount of variance of both strophiole area (ΔAIC = 34.953, ΔRMSE = 0.110) and strophiole relative size (ΔAIC = 13.977, ΔRMSE = 0.042).

## 3. Discussion

### 3.1. On the Taxonomic Value of Morphometric Seed Traits in Petrocoptis Seeds

The success of sexual reproduction and plant establishment, subsequent diversification and local adaptation ultimately rely on seeds [76]. Therefore, it is expected that seed-related traits may have undergone strong selective pressure and may keep a high phylogenetic signal. In fact, seed morphology has been used for classification purposes across several genera of Caryophyllaceae, where the taxonomic significance of vegetative and floral structures is scarce [38,77,78,79,80,81,82]. The presence of a strophiole separates *Petrocoptis* from its sister genera, and intraspecific classifications within the genus have always relied on seed morphology [38,39,71]. In this section, we focus on the taxonomic value of the studied seed traits, but other morphological traits may also have complementary taxonomic value.

Despite the high intra- and inter-population variability found in the measured *Petrocoptis* seed traits, it was possible to arrange an overall coherent set of three morphological groups implementing the K-means clustering algorithm. It is remarkable that most of the 11 taxa included in Montserrat & Fernández-Casas [39], a widely used taxonomic treatment of the genus, are coherently captured within any of the three groups: *P. crassifolia* in cluster “a” (●), *P. grandiflora*, *P. pseudoviscosa, P. pyrenaica* subsp. *pyrenaica* and *P. pyrenaica* subsp. *viscosa* in cluster “b” (▲) and *P. guarensis*, *P. hispanica* and *P. pardoi* in cluster “c”(■). Among the 20 *P. pyrenaica* subsp. *glaucifolia* populations analysed, only two—El Pindal (PIND) and Potes (POTE), which are geographically close to each other (Figure 6)—were classified in a different cluster. For *P. montsicciana*, the two northernmost populations—Beranuy (BERA) and Teller (TELL)—are recovered in our analysis apart from the rest and arranged in distinct clusters. The position of *P. montserratii* within the cluster arrangement remains unclear, as the three populations studied were split into two distinct clusters. Nevertheless, Figure 1 shows that these three populations occupy a nearby space in the ordination, indicating that they may share common morphological features, somehow intermediate among the three recovered clusters, although they have been separated by the K-means clustering algorithm. Summarizing, the resulting clustering of populations means, according to measured seed morphometric traits, resembles the taxonomic backbone selected as reference [39] in a synthetic way, so a given seed will be classified in the same cluster as all seeds belonging to the same species or subspecies. However, no differences were observed in the quantitative study of the seed traits that allow an analytical separation of all nine species (or eleven taxa, including the three subspecies, Table 4).

Walters [38] and Mayol & Rosselló [71,75] proposed more synthetic taxonomic classifications, as did our seed morphology analysis (Figure 1). Nonetheless, some of our results are not congruent with the specific and infraspecific arrangements proposed by these authors. The case of *P. hispanica* is notable because both Walters [38] and Mayol & Roselló [71,75] grouped this species with taxa included herein in cluster “b” (▲), like *P. pseudoviscosa* and *P. pyrenaica* (low seed area, low strophiole area, medium strophiole relative size), whereas our classification (based only on seed morphology) placed it closer to the species included in cluster “c” (■) (medium seed area, high strophiole area, high strophiole relative size). Similarly, our analysis does not support any difference in seed traits between *P. grandiflora* and *P. pyrenaica,* clearly differentiated by both Walters [38] and Mayol & Rosselló [71,75].

Further molecular work is required to trace these seed traits on an extensive phylogeny and see if they are congruent with it or, instead, if they are a case of adaptive convergence. Studies of other morphological and molecular characters are being conducted to try to propose a taxonomic treatment for the genus based on global evidence.

### 3.2. Relationship Between Climate and Morphological Traits of Petrocoptis Seeds

This is the first comprehensive study on patterns of seed traits and climate in *Petrocoptis*. Although there are some studies that explore possible relationships between climate-related variability and the morphology of plant diaspora in the literature [83,84,85,86,87], to our knowledge, no previous study has used data from representative natural populations collected throughout the entire distribution range of a chasmophytic genus.

*Petrocoptis* species live in limestone cliffs with some degree of temporal drought. Therefore, water availability in these populations varies depending primarily on local climate conditions, such as precipitation (directly or indirectly through infiltration) and evaporation linked to high temperatures. It has been described that seed size may increase drought tolerance [88], but this pattern does not fit the case of *Petrocoptis* at a climatic mesoscale level. The seed area of *Petrocoptis* showed no association with any of the climatic variables considered; thus, no evidence was found directly supporting the idea that the high intra- and interpopulation variability displayed by this trait is influenced by current or recent local climate. Instead, this variability could likely be explained by other factors, such as the existence of kinship networks that regulate seed size through maternal effects [89,90,91] or rock surface topographic heterogeneity, which affect microclimate components and, thus, resource availability at a microscale [3]. This can directly influence the vigour of individual plants growing in exceptionally favourable spots or that of an entire population (in case it grows in a particularly favourable cliff) and, thus, the size of the resulting mature seeds. Likewise, populations may be genetically distinct for this trait due to genetic drift or local adaptations [46,47,48].

Although it would appear reasonable to assume that the strophiole variability within *Petrocoptis* has taxonomic significance, ecological aspects should not be disregarded. The analysis of the strophiole identified a general and ubiquitous trend of differentiation of this structure throughout the entire distribution range and climatic spectrum of the genus. Even though Minuto et al. [82] hypothesized a correlation between the habitat in which the species grows and strophiole morphology in *Moehringia* L. (another strophiole-bearing Caryophyllaceae), similar relationships have not been previously explored when studying the strophiole or other analogous seed structures in Angiosperms. This strong trait–environment relationship opens the possibility that strophiole size variation has likely been driven by adaptation to local climate variables. Both raw strophiole area and its relative size exhibit clinal variation from cool/wet to warm/dry climates. Strophioles grown under low precipitation and/or warm temperature conditions are significantly bigger than those found in plants from high precipitation and/or cool temperature areas. Furthermore, the same trend is found when relative size is calculated (i.e., controlled the allometric effect that the seed size could have on it). This means that considering two individuals of the same seed size, the one growing under lower precipitation/warmer temperature conditions invests more resources in developing its strophiole than the one growing under higher precipitation/colder temperature. The ability of the strophiole to capture water helps the seed to hydrate in the overhanging cliffs with low water availability throughout the year. Our results point to an adaptive response through the investment of more resources in the production of bigger strophioles under dry and warm climates, increasing the hydration ability of the seed. Most populations and species integrating cluster “c” (■) —*P. pardoi*, *P. montsicciana* and *P. guarensis*—grow under Mediterranean macrobioclimate conditions [42], whereas those integrating cluster “b” (▲)—particularly *P. pyrenaica* and *P. grandiflora*—are more influenced by the temperate oceanic bioclimate (see below). However, *Petrocoptis* species present very marked geographic distribution patterns that strongly overlap with climatic gradients, which may lead to misinterpretation of the significant relationship found here between strophiole size and climate. Consequently, further research is needed (i.e., common garden experiments simulating climatic gradients) to test this hypothesis and clarify whether these relationships are or are not spurious.

Moreover, in the case that the effect of climate on the strophiole is confirmed, we are aware that it can be either a plastic- or allelic-dependent response. For perennial plants growing in regions showing important inter-annual climatic differences (as is the case of *Petrocoptis* in north Iberian mountains), it does not seem that allele selection through generation is the only adaptive pressure operating regarding climate, as it may take many years to modify allele frequencies and not every year the pressure may have the same intensity and direction. Common garden experiments simulating climatic gradients may help to clarify this issue, as they allow to test to what extent seeds of different origins present a fixed strophiole morphology independently of the climate (low plasticity and high genetically determined trait) or a variable behaviour according to climate (high plasticity and low genetically determined trait).

### 3.3. On the Biological Role of the Strophiole in Petrocoptis

Our results support the hypothesis that the strophiole is involved in the regulation of seed water balance. This outgrowth of the hilum region is capable of catching and retaining water so that it could be subsequently supplied to the seed [69]. Strophioles can regulate water movements in response to changes in external moisture but in an opposite way as compared to the hilum. While the hilum can open and close repeatedly in correspondence, respectively, with low and high environmental humidity, the strophiole remains closed at very low humidity [92,93,94]. Furthermore, variations in seed hydration may result from simultaneous fluctuations in temperature and humidity, which is likely what causes strophiole rupture in the field [94]. This is relevant on the face of the often overhanging limestone cliffs where *Petrocoptis* lives, a potential water-limited habitat [3] where fast and efficient seed hydration would improve germination and plant establishment. Furthermore, it appears to be critical for populations living under Mediterranean macrobioclimate conditions characterized by seasonal precipitation and summer water scarcity. In this context, it is likely that in the course of the evolution of the genus, the hygroscopic ability of the strophiole and its adaptation to water unavailability have secondarily favoured adhesion to the vertical surface of the rock and establishment in crevices where the soil is scarce or non-existent, as some researchers have already observed [72,74]. Following this hypothesis, this character could be understood as an exaptation [95]. The establishment of *Petrocoptis* in the rocky outcrops has probably benefited from these strophiole properties, along with the ability of the flower stems to reorient themselves towards the wall once the fruit has ripened, favouring the introduction of the seeds into the nearby cracks (i.e., geoautochory, active geocarpy according to [96,97,98]). Probably, apart from geoautochory, gravity (barochory), ombrohydrochory (rain-operated seed dispersal [99]) and even the intervention of certain ants, specialized or not, also act in *Petrocoptis* species. Under these premises, it is likely that *Petrocoptis* strophiole provides other adaptive advantages in addition to those discussed in this article.

The potential function of the strophiole in *Petrocoptis* as an ant attraction and reward structure needs to be discussed. Following the enunciation of the myrmecochory syndrome [64], it has recently been generalized for European and Iberian species that the presence of excrescences in the plant diaspora (i.e., elaiosomes) has a meaning related to its dispersal via ants [66,67]. The elaiosome is a lipid-rich appendage that can vary greatly in form, colour, hardness and size that can be found on seeds, fruits and other angiosperm organs [64,100]. Therefore, it brings together a broad spectrum of analogous structures (e.g., strophiole, aril) observed throughout different organs that share being mainly composed of lipids [67]. However, this characteristic has not been observed nor tested in the strophiolar tuft of *Petrocoptis*; neither has the case of the dispersal potential of ants in the case of *Petrocoptis* seeds. While it is not a focus of this study, exploring the relationship between *Petrocoptis* strophiole and ant-mediated dispersal will be part of our future research objectives.

This study is the first to interpret the seed strophiole of *Petrocoptis* as a potentially climate-related water uptake structure, which is especially important under adverse (dry) climatic conditions. Further physiological studies may help to deeply understand how this physical process occurs and how water uptake is regulated.

## 4. Materials and Methods

### 4.1. Plant Material: The Genus Petrocoptis

*Petrocoptis* is composed of chasmophytic chamaephytes endemic to the northern Iberian Peninsula. They live in the crevices of overhanging limestone cliffs, which are harsh environments for plant life. Both the delimitation of the genus and the number of accepted infrageneric taxa have changed over the last century (between four and twelve species, depending on the author), as studies of morphological traits [39,71,75,101,102,103] and both biochemical (isozymes; [104]) and genetic markers analyses (*ITS, rps16* intron; [105]) were carried out. Some of the traits that have traditionally been shown to be taxonomically important in the genus *Petrocoptis*, as well as other taxa within the family Caryophyllaceae, are found in the seed. It has a subreniform appearance and a large hygroscopic strophiolar tuft made of hairs that conceal an opening in the testa.

Based exclusively on practical reasons and in order to facilitate communication, in the first part of this study, we followed an easily available taxonomic treatment (included within the reference work *Flora iberica* http://www.floraiberica.es/, accessed on 10 November 2024), proposed by Montserrat & Fernández-Casas [39], in which the infrageneric classification of *Petrocoptis* consists of 9 species: *P. crassifolia* Rouy, *P. grandiflora* Rothm., *P. guarensis* Fern. Casas, *P. hispanica* (Willk.) Pau, *P. montserratii* Fern. Casas, *P. montsicciana* O. Bolòs & Rivas Mart., *P. pardoi* Pau, *P. pseudoviscosa* Fern. Casas and *P. pyrenaica*. Within the latter, 3 subspecies are recognized: *P. pyrenaica* subsp. *glaucifolia* (Lag.) P. Monts. & Fern. Casas, *P. pyrenaica* subsp. *pyrenaica* (Bergeret) A. Braun ex Walp. and *P. pyrenaica* subsp. *viscosa* (Rothm.) P. Monts. & Fern. Casas.

### 4.2. Sampling

Based on intensive field and herbarium investigations, 557 individuals from 84 populations of *Petrocoptis* were selected, and germplasm material was collected and preserved in silica gel. These populations are evenly distributed throughout the entire distribution range of the genus, and they cover both its whole altitudinal gradient and climatic spectrum (Figure 6). A total of 81 of the 84 populations were located in Spain, and 3 populations occurred on the French Pyrenees. Population information and specimen voucher numbers are listed in Appendix A. The voucher specimens were deposited in the herbarium of the Pyrenean Institute of Ecology-CSIC (JACA, herbarium codes following standard abbreviations from [106,107]). In each of the 84 populations, a minimum of three individuals were randomly selected for morphometric measurements. When more than three individuals had ripened dehiscent capsular fruit, the sampling was extended to a maximum of ten individuals per population. Conversely, the underrepresented populations were completed with previously collected herbarium material from BC, BIO, JACA, LEB, MA, SALA, SANT, VAL and VIT collections.

### 4.3. Measurement and Analysis of Morphological Traits

In total, 2773 seeds from 557 individuals were processed. Nine quantitative seed traits were selected following previous taxonomic accounts available for *Petrocoptis* [39] and other related taxa of the tribe Sileneae [108,109]. Seed digital images were captured with a Leica MC190 camera connected to an HD NIKON SMZ800 stereo microscope at 10× magnification with a resolution of 171 pixels mm^−1^. Then, images were semi-automatically processed with the software Leica Application Suite v4.12.0: seed and strophiole diameters were measured using software manual tools, and their respective area, roundness and strophiole relative size were automatically calculated (Figure 7).

For further clustering analysis, each population was considered as an operational taxonomic unit (OTU), and a data matrix was built using population mean values of all 9 morphological traits. We selected k-means as the cluster analysis technique to arrange populations based on seed morphology, as it is the most commonly used unsupervised machine learning algorithm for partitioning a given data set [110,111]. We applied a square root transformation to variables seed area and strophiole area and then we scaled/standardized the data to avoid the clustering algorithm dependence on an arbitrary variable unit. Euclidean distance was selected as a measurement of dissimilarity between observations. The NbClust R package (version 3.0.1) [112] provided 30 indices that helped determine the number of clusters (k) that will be generated in the final solution. Finally, the best k-means algorithm (k = 3; see Section 2) was carried out using the k-means function implemented in the stats R package and displayed with package factoextra (version 1.0.7) [113].

Then, we assessed the morphological recognisability of the three groups (those identified by the best k-means algorithm) in order to allow their identification and description. Thus, we estimated the normal fitted density distributions of 9 quantitative morphological traits for the 3 groups to assess the discontinuity and overlap of the variation of these morphological traits. One-way Welch’s ANOVA was performed to test the statistical significance of morphological differences among the groups proposed by k-means algorithm, and Games–Howell test was selected for pairwise comparisons. Both analyses were computed and displayed using ggstatsplot R package (version 0.12.2) [114]. Then, the coherence and predictability of each group were examined following Breiman’s random forest algorithm for classification [115,116], using randomForest R package (version 4.7.1.1) [117]. Then, we compared our results with the traditional taxonomic groups proposed by Montserrat & Fernández-Casas [39], Walters [38] and Mayol & Rosselló [71,75]. Although there are other taxonomic treatments available [101,102,118], they do not differ substantially from these ones. In order to test the predictive and classifying capacity of the model, it was trained with a subset of the raw seed data (70%) and later validated with the rest of them (30%). Given the high correlation between the length and width with the area of both the seed and strophiole, only the areas were included. In this way, the most explanatory variables were selected, discarding the redundant ones. The models were evaluated taking into account their confusion matrices, calculated using caret R package (version 6.0.94) [119] and the following general statistics:

(1) accuracy (Equation (1));
Accuracy = (True Positive + False Positive)/(True Positive + False Positive + True Negative + False Negative)(1)

(2) No information rate (i.e., how often the model would be wrong if it always predicted the majority class); 

(3) Cohen’s Kappa: a measure of how well the classifier performed as compared to how well it would have randomly performed [120,121]. The following specific statistics for each class (i.e., the three different clusters) were also explored:

(4) Sensitivity (Equation (2)):Sensitivity = (True Positive)/(True Positive + False Negative)(2)

(5) Specificity (Equation (3)):Specificity = (True Negative)/(False Positive + True Negative)(3)

(6) Balanced accuracy (Equation (4)):Balanced Accuracy = (Sensitivity + Specificity)/2(4)

Finally, the classificatory importance of each of the predictor variables (i.e., the diagnostic ability of each of them) was evaluated using the vip R package (version 0.4.1) [122]. The probability prediction of classification in a given cluster was explored following a support vector machine approach (SVM), implemented in e1071 R package (version 1.7.14) [123], selecting a radial basis kernel and setting C and gamma (γ) hyperparameters as default (C = 1, γ = 1/data dimension). The results were displayed as partial dependence plots and implemented in the pdp R package (version 0.8.1) [124] on a probability scale.

### 4.4. Identification of Relationships Among Seed Traits and Climatic Variables

We explored the relationship between climate and the seed traits and the following three results of interest: strophiole area, seed area and the ratio established between them. Linear mixed effects models (lme) were fitted between the traits and a set of climatic variables in order to determine the amount of variability explained by the environment. For this analysis, all 2773 *Petrocoptis* seeds were used as samples, and the genus’ perspective was followed as the reference taxonomic unit. Climate data were extracted for each population location from the Climatic Digital Atlas of the Iberian Peninsula [125], with a spatial resolution of 200 m. The following variables were considered as fixed effects: average annual rainfall, mean annual temperature, average annual maximum and minimum temperatures and average annual solar radiation. Furthermore, as observations were not independent nor exhaustive, individual and population levels were nested and included as random effects. We standardized all predictor variables to be comparable (i.e., equally weighted within the model), with a mean value of 0 and a standard deviation of 1, and seed trait values were transformed using square roots in order to achieve a normal fitted distribution of residuals. Then, linear mixed-effects models were performed to evaluate which predictor feature proposed a priori explained most of the variability. Each model, from the full one (with all the variables) to the null one (without any of them), was compared following the Akaike Information Criterion (AIC; [126,127]. Finally, after choosing the best one, we performed a backward model selection, starting with the best model and generating new simpler models by removing one variable at a time to determine the amount of variability of the seed traits that each could explain. The importance of each explanatory variable was evaluated by the calculation of a loss function that quantified the root-mean-squared-error (RMSE) goodness-of-fit measure drop of a given model after removing one variable at a time. It allowed us to compare the dependence of the models on climatic variables to make accurate predictions and to quantify the importance of random effects corresponding to the individual and population levels. This approach was performed by applying the feature_importance function on a previously created model explainer (explain function), setting 50 as the desired number of permutations. Algorithms were implemented in DALEX (version 2.4.3) and ingredients (version 2.3.0) R packages [128,129].

Finally, the K means grouping factor was added to the best model as a fixed effect in order to compare the amount of variability explained by cluster arrangement and climatic variables and to explore the relationship between seed morphology and climate across morphogroups.

## 5. Conclusions

The combination of seed morphometric traits of *Petrocoptis* alone allows the division of the genus into three well-defined morphogroups with known seed characteristics: one made up of large seeds (>2 mm^2^) with low strophiole relative size (<0.5), a second one composed by small seeds < 1.75 mm^2^) with small strophioles (<1 mm^2^) and a third one that includes seeds of high strophiole relative size (>1). Most of the taxa included in the taxonomic treatment that was used as reference [39] are coherently captured within any of the three groups, although no differences were observed in the quantitative study that allowed an analytical separation of all nine species (or eleven taxa, including the three subspecies proposed under *P. pyrenaica*). However, our results suggest that part of the intra- and interpopulation variability found in strophiole raw and relative size can be explained by climatic variables. We identified a general and ubiquitous trend of differentiation of this structure throughout the entire distribution range and climatic spectrum of the genus. This strong trait–environment relationship opens the possibility that strophiole size variation has likely been driven by adaptation to local climatic variables. Both raw strophiole area and its relative size exhibit clinal variation from cool/wet to warm/dry climates. This fact points to an adaptive response through the investment of more resources in the production of bigger strophioles in dry and warm climates, therefore increasing the hydration ability of the seed under these conditions. Similar relationships have not been previously explored when studying the strophiole or other analogous seed structures in Angiosperms. Our results lead us to reinterpret strophiole, in this case, as a climate-related structure involved in the regulation of seed water balance. It is likely that the adaptation of the strophiole to water unavailability, together with its hygroscopicity, secondarily favoured plant establishment and survival of *Petrocoptis* species in the crevices of vertical limestone cliffs throughout the course of the evolution of the genus.

## Figures and Tables

**Figure 1 plants-13-03208-f001:**
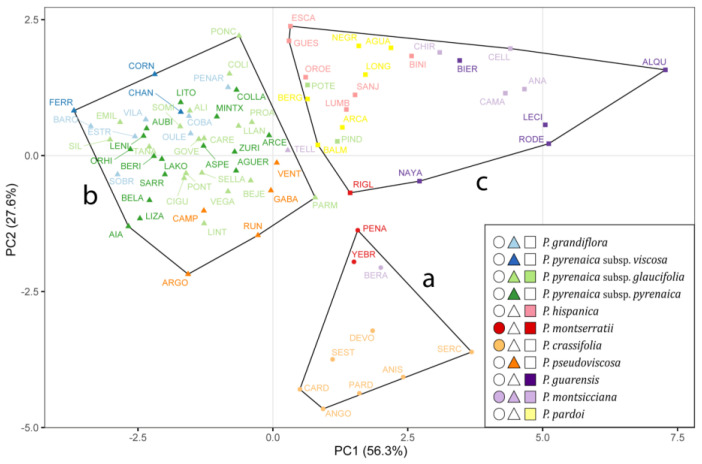
Cluster grouping of 84 populations of *Petrocoptis* A. Braun ex Endl. following k-means algorithm based on 9 morphological seed traits. Colours depict prior identification following the taxonomic proposal by Montserrat & Fernández-Casas [39], and symbol shapes indicate seed morphogroups: circles, cluster a; triangles, cluster b; squares, cluster c.

**Figure 2 plants-13-03208-f002:**
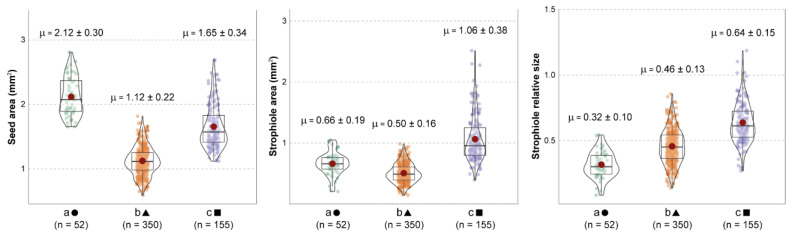
Comparison among the three k-means morphogroups (a, b and c), indicating the distribution, mean values and standard deviation of seed area (**left**), strophiole area (**center**) and strophiole relative size (**right**). Red dots indicate mean values and grey bars indicate median values. Every Games–Howell pairwise test showed significant differences.

**Figure 3 plants-13-03208-f003:**
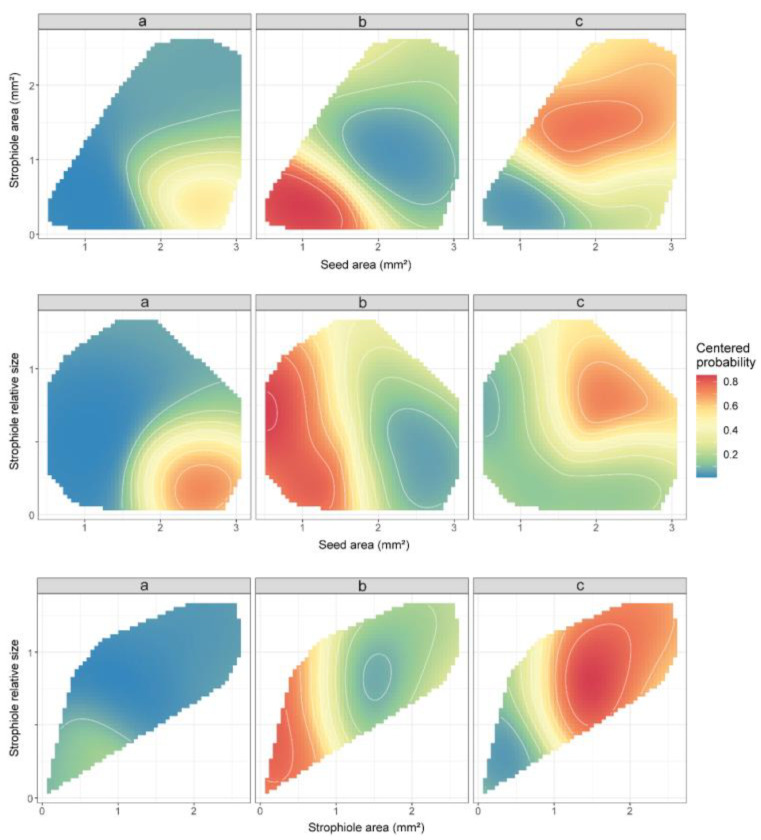
Partial dependence plots of the three morphogroups (a, b and c), based on seed area, strophiole area and strophiole relative size for the seed morphology data, derived from support vector machine (svm) algorithm. Colours depict the predicted classification probabilities of a *Petrocoptis* seed within a given group based on its morphological traits of interest. C and gamma hyperparameters were set as default: C = 1; γ = 1/(data dimension). Values are restricted to lie within the convex hull of their training values in order to avoid extrapolation.

**Figure 4 plants-13-03208-f004:**
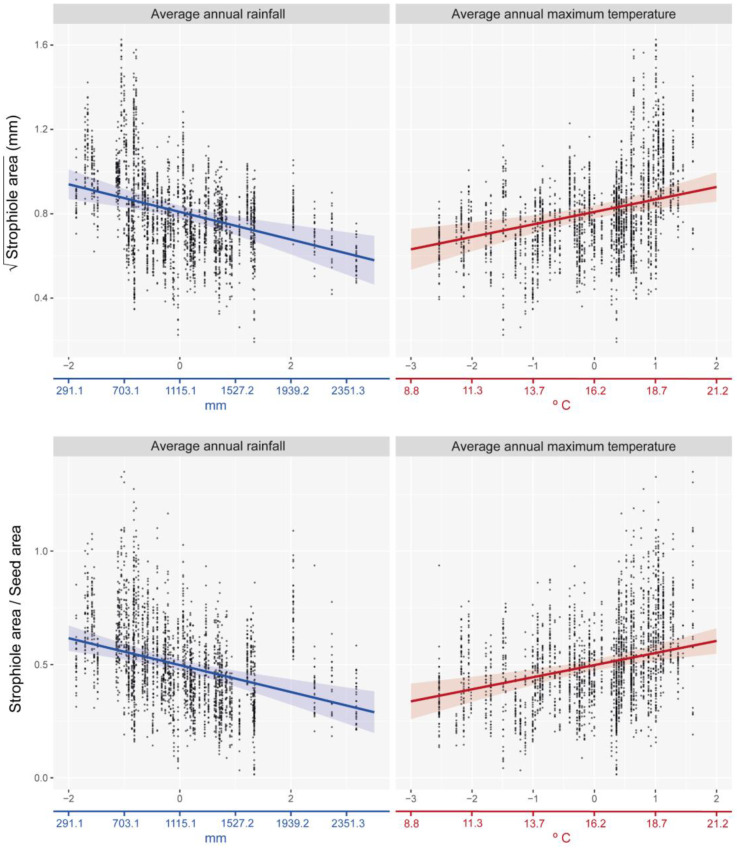
Distribution of observed data values and best-fitted linear mixed-effects models (lme) between climate variables and strophiole area (**up**) and strophiole relative size (**down**). Coloured X-axis is adjusted to show its original scale (prior standardization) for illustrative purposes.

**Figure 5 plants-13-03208-f005:**
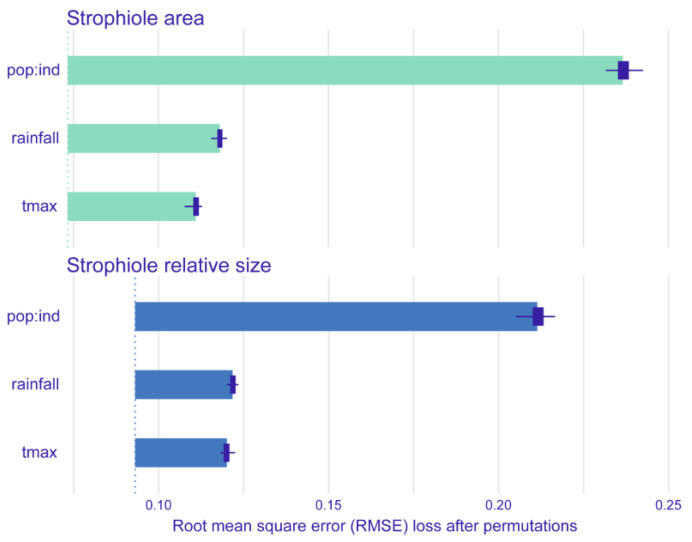
Mean variable importance of best-fitted linear mixed-effects models of strophiole area (**up**) and strophiole relative size (**down**). Dotted lines indicate RMSE value of the full models, and bars depict RMSE loss when removing one variable at a time. RMSE loss was calculated after 50 permutations. Variables included: nested population and individual levels (pop:ind), average annual rainfall (rainfall) and average annual maximum temperature (tmax).

**Figure 6 plants-13-03208-f006:**
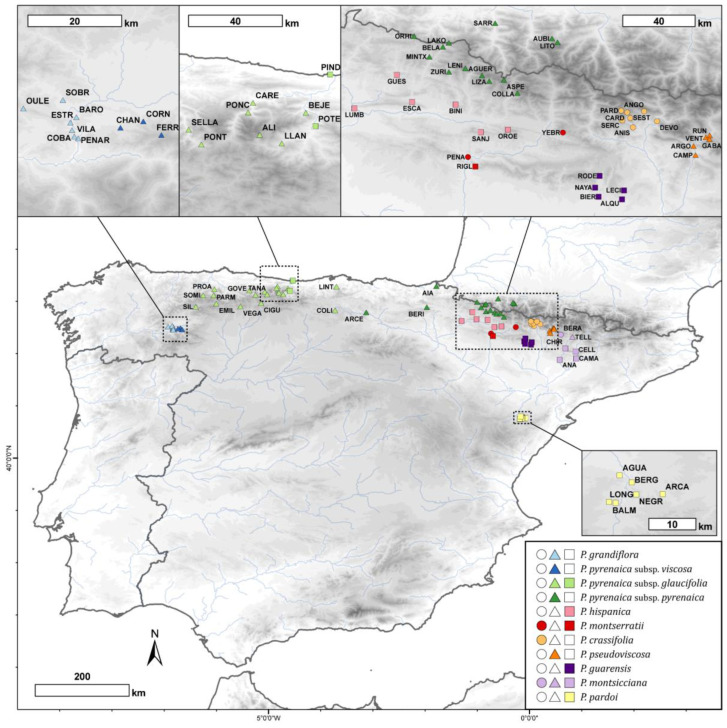
Geographic distribution of the *Petrocoptis* sampled populations. Symbol shapes indicate seed morphogroups (see Section 2 and Figure 1: circles, cluster a; triangles, cluster b; squares, cluster c) and colours depict prior identification following the taxonomic proposal by Montserrat & Fernández Casas [39]. Population codes follow Appendix A.

**Figure 7 plants-13-03208-f007:**
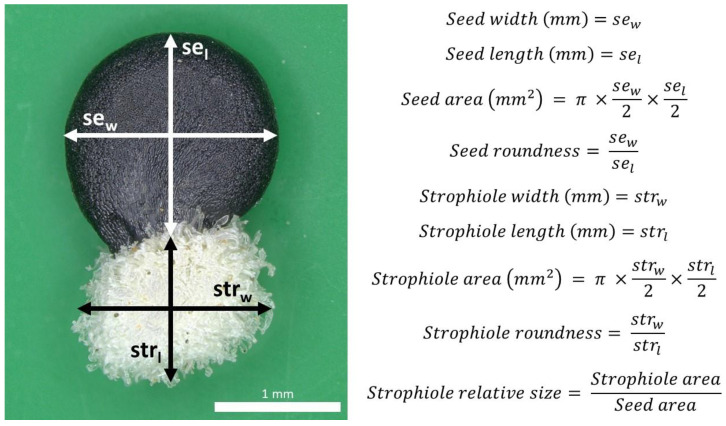
*Petrocoptis* seed traits of interest.

**Table 1 plants-13-03208-t001:** Confusion matrix (**left**) and statistics (**right**) of the model-trained random forest classification.

		Reference			Accuracy	0.8723		
		a	b	c	95% CI	(0.8475, 0.8943)		
Prediction	a	74	2	8	No Information Rate	0.6107		
	b	5	460	41	*p*-value [Acc > NIR]	<2 × 10^−16^		
	c	9	40	183	Kappa	0.7606		
					Statistics by class	a	b	c
					Sensitivity	0.8409	0.9163	0.7888
					Specificity	0.9864	0.8562	0.9169
					Balanced Accuracy	0.9136	0.8863	0.8529

**Table 2 plants-13-03208-t002:** Backward ∆AIC model selection of both strophiole area (**a**) and strophiole relative size (**b**). Average annual rainfall (rainfall) and average annual maximum temperature (tmax) were included as fixed effects in all models. For the best model (M_best_), null model (M_null_) and partial models (M_1_, M_2_), the following parameters are provided: Akaike Information Criterion (AIC_i_) values, AIC differences between M_best_ and M_1_, M_2_ and M_null_, degrees of freedom (df) and the marginal coefficient of determination (R^2^ marg.).

**(a)**						
**Model_i_**	**Variables**		**df**	**AIC_i_**	**ΔAIC_best-i_**	**R^2^ (marg.)**
M_best_	rainfall	tmax	6	−4721.343		0.627
M_1_	rainfall		5	−4716.826	4.517	0.547
M_2_		tmax	5	−4714.916	6.427	0.514
M_null_			4	−4700.129	21.214	0.000
**(b)**						
**Model_i_**	**Variables**		**df**	**AIC_i_**	**ΔAIC_best-i_**	**R^2^ (marg.)**
M_best_	rainfall	tmax	6	−5545.624		0.461
M_1_	rainfall		5	−5539.669	7.019	0.376
M_2_		tmax	5	−5538.395	7.334	0.330
M_null_			4	−5521.420	25.378	0.000

**Table 3 plants-13-03208-t003:** Summary of the best-supported models of strophiole area (**a**) and strophiole relative size (**b**). Each model includes average annual rainfall (rainfall) and average annual maximum temperature (tmax) as fixed effects. Variability at the population (pop) and individual (ind) levels is included as random effects. For each climate predictor and intercept, the following parameters are provided: coefficient estimates (Coefficient) with indication of the 95% confidence interval (95% CI), standard error (SE), t-statistic (t), degrees of freedom (df) and statistical significance (*p*-value).

**(a)**							
	**Parameter**	**Coefficient**	**SE**	**95% CI**	**t**	**df**	***p* Value**
Fixed effects	(Intercept)	0.81	0.02	(0.78, 0.84)	53.41	2137	<0.001
	rainfall	−0.07	0.02	(−0.10, −0.03)	−4.02	78	<0.001
	tmax	0.06	0.02	(0.03, 0.09)	3.74	78	<0.001
Random effects	SD pop (Intercept)	0.13					
	SD pop:ind (Intercept)	0.08					
	SD pop:ind (Residual)	0.08					
**(b)**							
	**Parameter**	**Coefficient**	**SE**	**95% CI**	**t**	**df**	***p* value**
Fixed effects	(Intercept)	0.50	0.01	(0.47, 0.52)	40.79	2137	<0.001
	rainfall	−0.06	0.01	(−0.08, −0.03)	−4.22	78	<0.001
	tmax	0.05	0.01	(0.03, 0.08)	4.17	78	<0.001
Random effects	SD pop (Intercept)	0.10					
	SD pop:ind (Intercept)	0.09					
	SD pop:ind (Residual)	0.10					

**Table 4 plants-13-03208-t004:** Comparison of the main taxonomic treatments proposed for the genera *Petrocoptis* and *Silene* (subg. *Petrocoptis*) and the assignment to each of the three morphogroups (a ●, b▲ and c ■) proposed in the present study. The classification accuracy of our seed data set, obtained by each of the random forest models, is indicated.

	Montserrat & Fernández Casas (1990) [39]		Walters (1993) [38]		Mayol & Rosselló (1999, corr. 2000) [71,75]		Present Study
Taxa	*Petrocoptis grandiflora* Rothm.	*Petrocoptis grandiflora* Rothm.	*Petrocoptis grandiflora* Rothm.	*Petrocoptis grandiflora* Rothm.	*Silene laxipruinosa* Mayol & Rosselló	*Silene laxipruinosa* Mayol & Rosselló	b ▲
	*Petrocoptis pyrenaica* (Bergeret) A.Braun ex Walp.	*P. pyrenaica* subsp. *viscosa* (Rothm.) P.Monts. & Fern.Casas	*Petrocoptis pyrenaica* (Bergeret) A.Braun ex Walp.	*P. pyrenaica* subsp. *viscosa* (Rothm.) P.Monts. & Fern.Casas	*Silene pyrenaica* (Bergeret) Mayol & Rosselló	*S. pyrenaica* subsp. *pyrenaica*	b ▲
		*P. pyrenaica* subsp. *glaucifolia* (Lag.) P.Monts. & Fern.Casas		*P. pyrenaica* subsp. *glaucifolia* (Lag.) P.Monts. & Fern.Casas			b ▲ *
		*P. pyrenaica* subsp. *pyrenaica*		*P. pyrenaica* subsp. *pyrenaica*			b ▲
	*Petrocoptis hispanica* Pau	*Petrocoptis hispanica* Pau	*Petrocoptis hispanica* Pau	*Petrocoptis hispanica* Pau			c ■
	*Petrocoptis pseudoviscosa* Fern. Casas	*Petrocoptis pseudoviscosa* Fern. Casas				*S. pyrenaica* subsp. *pseudoviscosa* (Fern. Casas) Mayol & Rosselló	b ▲
	*Petrocoptis crassifolia* Rouy	*Petrocoptis crassifolia* Rouy	*Petrocoptis crassifolia* Rouy	*Petrocoptis crassifolia* Rouy	*Silene montserratii* (Fern. Casas) Mayol & Rosselló	*S. montserratii* subsp. *crassifolia* (Rouy) Mayol & Rosselló	a ●
	*Petrocoptis montserratii* Fern. Casas	*Petrocoptis montserratii* Fern. Casas				*S. montserratii* subsp. *montserratii*	a ● *
	*Petrocoptis guarensis* Fern. Casas	*Petrocoptis guarensis* Fern. Casas	*Petrocoptis pardoi* Pau	*Petrocoptis pardoi* Pau	*Silene pardoi* (Pau) Mayol & Rosselló	*S. pardoi* subsp. *guarensis* (Fern. Casas) Mayol & Rosselló	c ■
	*Petrocoptis montsicciana* O.Bolòs & Rivas Mart.	*Petrocoptis montsicciana* O.Bolòs & Rivas Mart.				*S. pardoi* subsp. *pardoi*	c ■ *
	*Petrocoptis pardoi* Pau	*Petrocoptis pardoi* Pau					c ■
N_groups_	9 (species)	11 (species + subspecies)	5 (species)	7 (species + subspecies)	4 (species)	7 (species + subspecies)	3 (morpho- groups)
Accuracy	63.14%	47.69%	68.49%	53.77%	76.89%	68.61%	87.23%

Asterisks (*) indicate that not all populations of the reference taxa [39] were grouped in the same cluster.

## Data Availability

The data presented in this study are contained within the article and Appendix A and openly available in Zenodo at https://doi.org/10.5281/zenodo.13972509.

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
