# Peer review of "Classification Importance of Seed Morphology and Insights on Large-Scale Climate-Driven Strophiole Size Changes in the Iberian Endemic Chasmophytic Genus Petrocoptis (Caryophyllaceae)"

_plants, 2024, doi:10.3390/plants13223208_

Round 1
Reviewer 1 Report
Comments and Suggestions for Authors
This article studied the biological role of strophiole of Petrocoptis and explored relationships between strophiole size and climates in order to understand adaptation and evolution of seed morphology such as strophiole sizes or relative sizes, to environment, and further apply to Petrocoptis classification with the found seed characteristics. The scientific question is novel and the conclusions are sound.
Minor details need to think or revised as following:
1) The title needs to be revised to fit the main idea of this article, “classification of Petrocoptis” may be better than “water uptake”.
2) Page4, line 59, 16 proposed three as the best number of clusters, unclear.
3) “Fig.1” and “Fig.6” are difficult to be understood, where is a, b and c? Colours depict taxonomic assignment following [39], you had better to explain them in detail here, instead of referring to [39].
Author Response
Summary:
We sincerely appreciate that you have taken the time to review this manuscript. Below, you’ll find our thoughtful responses, along with the revisions and corrections clearly highlighted in the resubmitted files. Thank you again for your valuable feedback.
Point-by-point response to Comments and Suggestions for Authors:
Comments 1: The title needs to be revised to fit the main idea of this article, “classification of Petrocoptis” may be better than “water uptake”.
Response 1: Thank you for pointing this out. We agree with this comment. Therefore, we have changed the title according to your suggestion, making it more clear and complete: Classification importance of seed morphology and insights on large scale climate-driven strophiole size changes in the Iberian endemic chasmophytic genus Petrocoptis (Caryophyllaceae).
Comments 2: Page4, line 59, 16 proposed three as the best number of clusters, unclear.
Response 2: Agreed and changed accordingly. We have rewritten the phrase to make it clearer:
Lines 157-160:
Sixteen out of the 27 indices calculated by the NbClust algorithm converged in k=3 as the optimal number of clusters. The remaining 11 indices showed much weaker support, with less than four of them agreeing on a different number of clusters (Table S2).
Comments 3: “Fig.1” and “Fig.6” are difficult to be understood, where is a, b and c? Colours depict taxonomic assignment following [39], you had better to explain them in detail here, instead of referring to [39].
Response 3: We agree. We have rewritten both captions to make them more straight-forward. Letters a, b & c are now depicted in the figure.
Page 4, lines 173-176:
Figure 1. Cluster grouping of 84 populations of Petrocoptis following k-means algorithm based on 9 morphological seed traits. Colours depict prior identification following the taxonomic proposal by Montserrat & Fernández-Casas [39] and symbol shapes indicate seed morphogroups: circles, cluster a; triangles, cluster b; squares, cluster c.
Page 10, lines 309-312:
Figure 6. Geographic distribution of the Petrocoptis sampled populations. Symbol shapes indicate seed morphogroups (see Results and Figure 1: circles, cluster a; triangles, cluster b; squares, cluster c) and colours depict prior identification following the taxonomic proposal by Montserrat & Fernández Casas [39]. Population codes follow Table S1.

Reviewer 2 Report
Comments and Suggestions for Authors
The manuscript “Insights on large scale climate-driven strophiole size changes related to water uptake in the Iberian endemic chasmophytic genus Petrocoptis (Caryophyllaceae)” by the authors: Jorge Calvo-Yuste, Ángela Lis Ruiz-Rodríguez, Brais Hermosilla, Agustí Agut, M. Montserrat Martínez-Ortega & Pablo Tejero deals with the morphology of the seeds of Petrocoptis to assess whether the measured morphometric seed traits can contribute to the description of seed morphogroups that hold taxonomic significance and to explore whether a relationship can be established between these traits and climate; and at least try to get biological role to the strophiole part of the seed in Petrocoptis genus.
The paper is well coordinated with functional elaborations to interpret the relationships between the seed traits and function and taxonomy contribution.
FOR THE AUTHORS
The authors declare that samples were collected throughout the entire distribution range of the genus, and they covered its entire altitudinal gradient.
I invite the authors to add the altitude of the stands to the table and to comment if there are relationships with differences in altitude
Line 32: “I suggest authors replace the keywords already present in the title of the paper
Line 44: change “ alpine orogeny” to “Alpine orogeny”
Line 81-82: change adding commas ”, from which it differs, among other traits, by the presence of a strophiole, a characteristic small tuft of hairs by the hilum of its seed
Line 96: change “masl” to “m a.s.l.”
Line 104-107: “Besides, it is probable that phenotypic plasticity and precipitation or
temperature are correlated, with benign conditions favoring greater phenotypic plas-
ticity [57,58] due to a wider range of morphological and physiological variations [59]. On
the contrary, stressful conditions may place restrictions on phenotypic plasticity [60)”
This sentence is not clear. the first sentence should specify which temperature or precipitation values could be positive for greater phenotypic plasticity because the same T and P values can be positive or negative based on the autoecology of the individual species and the reference habitat
Line 174: Figure 1
The authors are strongly invited to add the letters a,b, and c to the graph to identify the three groups.
Line 248: Table 2
The caption must be rewritten and include an explanation of all parameters. check if the parameters are written in the same manner both in the caption and table
Line 252: Table 3
The caption must be rewritten and include an explanation of all parameters
Line 313: Why in the table the black balls (subgroup a) are too large?
Author Response
Summary
We sincerely appreciate that you have taken the time to review this manuscript. Below, you’ll find our thoughtful responses, along with the revisions and corrections clearly highlighted in the resubmitted files. Thank you again for your valuable feedback.
Point-by-point response to Comments and Suggestions for Authors
Comments 1: The authors declare that samples were collected throughout the entire distribution range of the genus, and they covered its entire altitudinal gradient.
I invite the authors to add the altitude of the stands to the table and to comment if there are relationships with differences in altitude
Response 1: Thank you for pointing out this issue.
The altitude of each population is indicated under the "Locality" column of Table S1 (Supplementary Material).
The second part of this comment is really sharp!, thanks. In fact, we considered altitude as another good candidate for a predictor variable of seed morphology during the phase of tuning and selecting the best linear models. However, we discarded it when we were exploring the variables because it was strongly correlated with temperature (as expected! See the graph in the attached file). Therefore, we considered it redundant and out of our scope.
We believe that altitude itself is not important in this regard. What we consider biologically relevant is the effect it can have on other environmental, biotic or abiotic variables.
Comments 2: Line 32: I suggest authors replace the keywords already present in the title of the paper
Response 2: Agreed and changed. We have, accordingly, modified the keywords to avoid redundancies with the article title.
Lines 32-33:
Keywords: chasmophyte; cliff environment; Iberian Peninsula; LMEs; local adaptation; machine learning; morphometrics; random forest; water availability.
Comments 3: Line 44: change “alpine orogeny” to “Alpine orogeny”
Response 3: Agreed and changed accordingly.
Line 43: Alpine orogeny
Comments 4: Line 81-82: change adding commas ”, from which it differs, among other traits, by the presence of a strophiole, a characteristic small tuft of hairs by the hilum of its seed
Response 4: Agreed.
Line 80-81: ”from which it differs, among other traits, by the presence of a strophiole, a characteristic small tuft of hairs by the hilum of its seed”
Comments 5: Line 96: change “masl” to “m a.s.l.”
Response 5: Agreed.
Line 95: m a.s.l.
Comments 6: Line 104-107: “Besides, it is probable that phenotypic plasticity and precipitation or temperature are correlated, with benign conditions favoring greater phenotypic plasticity [57,58] due to a wider range of morphological and physiological variations [59]. On
the contrary, stressful conditions may place restrictions on phenotypic plasticity [60)”
This sentence is not clear. The first sentence should specify which temperature or precipitation values could be positive for greater phenotypic plasticity because the same T and P values can be positive or negative based on the autoecology of the individual species and the reference habitat.
Response 6: Thank you for bringing this up. Unfortunately, we have not yet explored this particular aspect, so we are unable to provide an answer at this time, nor to include it in the manuscript. Additionally, there are no existing studies that have addressed this in Petrocoptis, which is why the statement remains open.
Comments 7: Line 174: Figure 1: The authors are strongly invited to add the letters a, b, and c to the graph to identify the three groups.
Response 7: You are right. Letters a, b & c are now depicted in the figure.
Comments 8: Line 248: Table 2: The caption must be rewritten and include an explanation of all parameters. check if the parameters are written in the same manner both in the caption and table
Response 8: Agreed. We have, accordingly, modified the caption to properly explain all parameters.
Line 248-253: “Table 2. Backward ΔAIC model selection of both strophiole area (a) and strophiole relative size (b). Average annual rainfall (rainfall) and average annual maximum temperature (tmax) were included as fixed effects in all models. For the best model (Mbest), null model (Mnull), and partial models (M1, M2), the following parameters are provided: Akaike Information Criterion (AICi) values, AIC differences between Mbest and M1, M2, and Mnull, degrees of freedom (df), and the marginal coefficient of determination (R² marg.)”.
Comments 9: Line 252: Table 3: The caption must be rewritten and include an explanation of all parameters
Response 9: Thank you. Again, we modified the caption to make it more comprehensible.
Line 254-259:
“Table 3. Summary of the best-supported models of strophiole area (a) and strophiole relative size (b). Each model includes average annual rainfall (rainfall) and average annual maximum temperature (tmax) as fixed effects. Variability at the population (pop) and individual (ind) levels is included as random effects. For each climate predictor and intercept, the following parameters are provided: coefficient estimates (Coefficient) with indication of the95% confidence interval (95% CI), standard error (SE), t-statistic (t), degrees of freedom (df), and statistical significance (p-value)”.
Comments 10: Line 313: Why in the table the black balls (subgroup a) are too large?
Response 10: Line 317 Table 4: There was a problem with the default font type. The same font size (7) correspond with different real sizes depending on the shape. We have adjusted it and now shapes are more or less equally sized.
